# Cholesterol Modulation Attenuates the AD-like Phenotype Induced by Herpes Simplex Virus Type 1 Infection

**DOI:** 10.3390/biom14050603

**Published:** 2024-05-20

**Authors:** Blanca Salgado, Beatriz Izquierdo, Alba Zapata, Isabel Sastre, Henrike Kristen, Julia Terreros, Víctor Mejías, María J. Bullido, Jesús Aldudo

**Affiliations:** 1Centro de Biologia Molecular Severo Ochoa (CBM), CSIC-UAM, Universidad Autonoma de Madrid, 28049 Madrid, Spain; bsalgado@cbm.csic.es (B.S.); bea.izquierdoal@gmail.com (B.I.); albazapataalcazar97@gmail.com (A.Z.); isastre@cbm.csic.es (I.S.); henrikekristen@gmail.com (H.K.); julia.terrerosr@gmail.com (J.T.); pi.victormejias@gmail.com (V.M.); 2Centro de Investigacion Biomedica en Red Sobre Enfermedades Neurodegenerativas (CIBERNED), 28031 Madrid, Spain; 3Hospital Clinico San Carlos, 28040 Madrid, Spain; 4Institute for Bioengineering of Catalunya (IBEC), Barcelona Institute of Science and Technology (BIST), 08028 Barcelona, Spain; 5Institute for Research in Biomedicine (IRB Barcelona), Barcelona Institute of Science and Technology (BIST), 08028 Barcelona, Spain; 6Faculty of Medicine and Health Sciences, University of Barcelona, 08036 Barcelona, Spain; 7Hospital La Paz Institute for Health Research, IdiPAZ, 28046 Madrid, Spain

**Keywords:** Alzheimer’s disease, HSV-1, cholesterol, neuroblastoma cells, methyl-beta-cyclodextrin, infection, neurodegeneration, lysosomal alterations, beta-amyloid, hyperphosphorylated tau

## Abstract

Cholesterol, a crucial component of cell membranes, influences various biological processes, including membrane trafficking, signal transduction, and host-pathogen interactions. Disruptions in cholesterol homeostasis have been linked to congenital and acquired conditions, including neurodegenerative disorders such as Alzheimer’s disease (AD). Previous research from our group has demonstrated that herpes simplex virus type I (HSV-1) induces an AD-like phenotype in several cell models of infection. This study explores the interplay between cholesterol and HSV-1-induced neurodegeneration. The impact of cholesterol was determined by modulating its levels with methyl-beta-cyclodextrin (MβCD) using the neuroblastoma cell lines SK-N-MC and N2a. We have found that HSV-1 infection triggers the intracellular accumulation of cholesterol in structures resembling endolysosomal/autophagic compartments, a process reversible upon MβCD treatment. Moreover, MβCD exhibits inhibitory effects at various stages of HSV-1 infection, underscoring the importance of cellular cholesterol levels, not only in the viral entry process but also in subsequent post-entry stages. MβCD also alleviated several features of AD-like neurodegeneration induced by viral infection, including lysosomal impairment and intracellular accumulation of amyloid-beta peptide (Aβ) and phosphorylated tau. In conclusion, these findings highlight the connection between cholesterol, neurodegeneration, and HSV-1 infection, providing valuable insights into the underlying mechanisms of AD.

## 1. Introduction

The biological significance of cholesterol is given by its regulation of cell membrane structure and fluidity and its role as a precursor of diverse compounds, such as steroid hormones, oxysterol, and bile acids. Cholesterol is involved in a wide range of processes, including membrane trafficking, signal transduction, host-pathogen interactions, and immune function. Thus, the maintenance of cholesterol homeostasis is essential for the proper functioning of cells, and its disruption has been related to several human congenital (Nieman-Pick type C disease (NPC), familial hypercholesterolemia) and acquired (cardiovascular disorders, neurodegenerative processes, and certain types of cancer) diseases [1,2,3]. In this context, cellular cholesterol arises as a crossroad in cell pathophysiology, which could lead to the discovery of new molecular mechanisms involved in disease progression or even serve as therapeutic targets. 

Cholesterol is derived from both dietary sources and de novo biosynthesis, primarily occurring in the liver and the brain. The central nervous system (CNS) contains approximately 20–25% of total cholesterol, highlighting its crucial role in neuronal function. Most CNS cholesterol resides in myelin and is essential for synaptogenesis, synaptic transmission, and plasticity [2]. Cholesterol homeostasis and metabolism have been extensively studied in the field of neurodegenerative disorders such as Alzheimer’s disease (AD) (reviewed in [4])—the most common form of dementia. Although the main hallmarks of AD—senile plaques mainly composed of beta-amyloid (Aβ) peptides and neurofibrillary tangles composed of hyperphosphorylated tau protein—have been known for more than a century [5], the molecular mechanisms involved in disease progression remain obscure. Recently, cholesterol, along with other genetic and non-genetic factors, has emerged as a potential contributor to the sporadic forms of the disease [6]. 

On the other hand, previous reports have accounted for the ability of certain viruses to induce metabolic reprogramming in host cells and interfere with cholesterol homeostasis [7,8]. For many years, our lab has focused on studying the links between AD and herpes simplex virus type I (HSV-1) infection. HSV-1 is a highly prevalent DNA virus that, apart from infecting epithelial cells and causing mucosal lesions, can also reach the nervous system and establish latent infections [9]. We have characterized several in vitro models using different cell lines—murine and human neuroblastoma cells and human neural precursors—where AD hallmarks were reported upon HSV-1 infection. This phenotype includes the inhibition of Aβ secretion, intracellular accumulation of Aβ and hyperphosphorylated tau protein, and alterations in the autophagy-lysosomal pathway [10,11,12,13,14]. These results contribute to the growing evidence supporting the infectious hypothesis of AD and the potential role of HSV-1 in neurodegeneration, initially proposed at the end of the 20th century [15,16]. Recent studies point out that susceptibility to HSV-1 could be aggravated by the *APOE4* genotype, suggesting a shared risk factor for infection and AD neurodegeneration [17].

Genetic expression studies in our cell model of infection and oxidative stress identified the lysosomal pathway and cholesterol homeostasis as the main pathways altered [14]. Post-mitotic cells such as neurons are particularly sensitive to dysfunction of the lysosomal machinery. Defects in autophagy and endocytosis may thus play an important role in neurodegenerative diseases, including AD, as suggested by previous reports [18,19]. In line with this, we have described that HSV-1 infection is able to alter both the lysosomal load and lysosome activity. Moreover, we found that deficiency in LAMP2—a lysosome membrane protein involved in the fusion of autophagosomes with lysosomes and cholesterol trafficking—leads to a significant reduction of viral infection and attenuates the neurodegeneration markers induced by the virus [20].

Overall, we aim to keep deciphering the connections between HSV-1 infection and AD neurodegeneration. In this work, considering that cholesterol dysregulation has been associated with markers of AD [4] and proposed to play multiple roles in the HSV-1 replicative cycle [21], we focus on the potential contribution of cholesterol as a bridge between HSV-1 and AD neurodegeneration. Using an in vitro approach to modulate cholesterol levels in neuroblastoma cells, we have observed that HSV-1 triggers intracellular cholesterol accumulation. Moreover, lowering cholesterol levels mitigates infection and ameliorates certain aspects of neurodegeneration, including Aβ accumulation, tau hyperphosphorylation, and lysosomal alterations within infected cells. These results point to a pivotal role of cellular cholesterol in the AD-related features promoted by HSV-1 in neuronal cells. 

## 2. Materials and Methods

Cell lines and culture medium. SK-N-MC cells (HTB-10), initially described as neuroblastoma and afterward cataloged as part of the Ewing’s sarcoma tumor family [22], were obtained from the American Type Culture Collection (ATCC). They were cultured as a monolayer in minimal Eagle’s medium (MEM) supplemented with 10% heat-inactivated fetal calf serum, 1 mM sodium pyruvate, nonessential amino acids, 2 mM glutamine, and 50 μg/mL gentamycin. Murine neuroblastoma cell line N2a was supplied by Paul Saftig’s lab [23]. N2a cells were cultured in Dulbecco’s modified Eagle medium (DMEM) supplemented with 10% fetal calf serum, 2 mM glutamine, nonessential amino acids, 1 mM sodium pyruvate, and 50 μg/mL gentamycin. Both cell lines were cultured at 37 °C in a humidified 5% CO_2_ atmosphere.

HSV-1 infection. When cell cultures reached 70–75% confluence, they were infected with wild-type HSV-1 strain KOS 1.1 at a multiplicity of infection (MOI) of 10 plaque-forming units per cell (pfu/cell). The virus was obtained, propagated, and purified from Vero cells, as described in [24]. Two different procedures were followed. In the first procedure, cells were exposed to viral solution for 1 h at 37 °C. Subsequently, the unbound virus was removed, and the cells were further incubated in a culture medium at 37 °C until collection. In the second procedure, cells were incubated with the viral solution for 2 h at 37 °C. After viral adsorption, cells were treated for two minutes with citrate buffer pH 3 and then culture medium was added. Non-infected samples (mock) were incubated in virus-free suspensions. Viral titers in cell lysates (intracellular viral titer) and cell culture supernatants (extracellular viral titer) were determined by plaque assay. Finally, to modify cholesterol levels, 1.5–3 mM methyl-beta-cyclodextrin (MβCD), 25 μg/mL U18666A, and 15 μg/mL cholesterol were used (Sigma-Aldrich, St. Louis, MO, USA). In all experiments, the drug MβCD was added to the culture medium after viral adsorption, except in the assays intended to monitor virus entry, where it was added one hour before viral adsorption. MβCD was not in contact with the viral solution in any of the settings used.

Viral DNA quantification. QIAamp^®^ DNA Blood Kit (QIAGEN, Hilden, Germany) was used to purify DNA. For HSV-1 DNA quantification, real-time quantitative PCR was performed with a CFX-384 Real-Time PCR System (BioRad, Hercules, CA, USA) using a custom-designed TaqMan assay specifically for the *US12* viral gene (forward primer: 5′-CGTACGCGATGAGATCAATAAAAGG-3′; reverse primer: 5′-GCTCCGGGTCGTGCA-3′; TaqMan probe: 5′-AGGCGGCCAGAACC-3′). Viral DNA content was normalized, in terms of human genomic DNA, and quantified with a predesigned TaqMan assay specifically for the *18S* (Hs9999991_s1; Applied Bio-systems, Waltham, MA, USA). The quantification data were expressed as a viral DNA copy number per ng of genomic DNA. 

Cell viability assay. To assess cell viability upon different treatments, 3-(4,5-dimethylthiazol-2-yl)-2,5-diphenyltetrazolium bromide (MTT) reduction to formazan colorimetric assay was performed. Briefly, cells were cultured in 96-well plates and 0.5 mg/mL MTT (Sigma-Aldrich, St. Louis, MO, USA) was added. Two hours later, 100 μL of solubilization buffer (20% SDS, 50% N, N-dimethyl-formamide, pH 4.7) was added, followed by an overnight incubation at 37 °C, protected from light. The next day, formazan absorbance was measured at 550 nm in an ELISA Microplate Reader model 680 (BioRad, Hercules, CA, USA). 

Cholesterol quantification. Intracellular cholesterol levels were quantified using the enzymatic fluorogenic assay Amplex Red Cholesterol Assay Kit (Invitrogen, Waltham, MA, US), following the provider’s instructions. Since cholesterol esterase was not added to the assay reaction mix, we only considered free cholesterol for our measurements. For the experiments, 10 μg of protein lysates were analyzed. Fluorescence was measured in a plate fluorimeter at 560 nm excitation and 590 nm emission wavelengths (BMG LABTECH, Saitama, Japan).

Immunofluorescence analysis and filipin staining. After growing cells on coverslips, fixation in 4% paraformaldehyde (PFA) and permeabilization with blocking solution (2% horse or fetal calf serum, 0.2% Triton X-100 in phosphate buffer saline [PBS] pH 7.4) were performed. Incubations with primary antibodies and Alexa Fluor-coupled secondary antibodies were carried out in a blocking solution (Table 1). Finally, cells were counterstained with 4,6-diamino-2-phenylindole (DAPI) (Merck, Rahway, NJ, USA) in PBS and mounted on microscope slides using Mowiol medium (Sigma-Aldrich, St. Louis, MO, USA). The overall procedure took place at room temperature (RT) and samples were protected from light. 

For filipin staining, 4% PFA-fixed cells were incubated for 30 min with 50 μg/mL filipin (Sigma-Aldrich, St. Louis, MO, USA) and 1 μM TO-PRO-3 (Thermo Fisher, Waltham, MA, USA), a compound that binds to nuclear DNA, in PBS before mounting with Mowiol medium.

Sample visualization was performed in a FRET inverted microscope Axiovert200 (Zeiss, Jena, Germany) coupled to a monochrome ccd camera and in a LSM 900 laser scanning confocal microscope (Zeiss) coupled to a vertical Axio Imager 2 vertical microscope (Zeiss). Immunofluorescence images were obtained using Metamorph (Molecular Devices, San Jose, CA, USA) or ZEN Blue 3.4 imaging software (Zeiss, Jena, Germany) and processed with Adobe Photoshop (Adobe, San Jose, CA, USA).

Cell lysates and Western blot analysis. Lysis of cell cultures was achieved by incubating cell samples in a preparation containing proteases (CompleteTM, Mini, EDTA-free Protease Inhibitor Cocktail, Roche, Basel, Switzerland) and phosphatases inhibitors (PhosSTOPTM, Roche) in the radioimmunoprecipitation assay (RIPA) buffer (10 mM Tris-HCl pH 7.5, 50 mM NaCl, 1% Nonidet P-40, 0.2% sodium deoxycholate, 0.1% sodium dodecyl sulfate [SDS], 1 mM EDTA). Before Western blot analysis, protein concentration in cell lysates was determined by Bicinchoninic acid assay (BCA, Pierce, Waltham, MA, USA), following the provider’s instructions. Then, Laemmli discontinuous SDS-polyacrylamide gel electrophoresis was performed. After electrophoresis and transfer to a nitrocellulose membrane, membranes were blocked with PBS-3% BSA-0.2% Tween 20 or PBS-5% nonfat milk-0.2% Tween 20 solution. Incubations with primary and secondary antibodies coupled to peroxidase (Table 1) were carried out in dilution buffer (PBS-1% BSA-0.05%Tween 20 or PBS-1% nonfat milk -0.05% Tween 20) at RT. Washing steps were performed in PBS-0.05% Tween 20. Finally, detection by enhanced chemiluminescence was carried out using ECL Western blotting detection reagents (Amersham Biosciences, Amersham, UK) according to the manufacturer’s instructions.

Quantification of lysosome load. The lysosome load was determined using the acidotropic probe LysoTracker Red DND-99 (LTR, Thermo Fisher, Waltham, MA, USA), which accumulates in acidic organelles. To assure assay reliability, cell exposure to Bafilomycin A1 (0.1 μM) (Sigma-Aldrich, St. Louis, MO, USA) was performed. Before the end of infection or treatments, cells were exposed to 0.5 μM LTR for 1 h at 37 °C in a culture medium and then washed with PBS. Then, cells were lysed with RIPA buffer and centrifuged at 13,000× *g* for 10 min. The BCA method was used to estimate protein concentration in lysates, and fluorescence was recorded using a FLUOstar OPTIMA microplate reader (BMG LABTECH, Saitama, Japan) (excitation wavelength 560 nm and emission wavelength 590 nm). 

Cathepsin activity assays. Cathepsin activity assays were conducted as detailed by [25] with minor modifications. Briefly, cells were lysed in a buffer containing 50 mM sodium acetate (pH 5.5), 0.1 M NaCl, 1 mM EDTA, and 0.2% Triton X-100. Lysates were clarified by centrifugation and immediately used for the determination of proteolytic activity. Then, 20–50 μg of protein from cell lysates were incubated for 30 min in the presence of the following fluorogenic substrates (all from Enzo Life Sciences, Farmingdale, NY, USA): Z-VVR-fluorophore 7-amino-4-methyl-coumarin (AMC) (P-199; most sensitive substrate for cathepsin S; 20 mM) and the Cathepsin D/E fluorogenic substrate Mca-GKPILFFRLK (Dnp)-D-Arg-NH2 (P-145; 10 mM). Measurements were taken in a microtiter plate reader (Tecan Trading AG, Männedorf, Switzerland) at excitation/emission 360/430 nm or 320/400 nm for Z-VVR-AMC and Cathepsin D and E fluorogenic substrates, respectively. 

Secreted Aβ Measurements. Conditioned media from non-infected and infected cultures were assayed for Aβ40 and Aβ42 using commercial sandwich enzyme-linked immunosorbent assay (ELISA) kits (Wako, Tokyo, Japan) according to the manufacturer’s instructions. First, inactivation of collected media was performed through UV exposure. After centrifugation, samples were kept at −70 °C prior to 5-fold concentration by lyophilization and suspension in PBS with protease inhibitor cocktail (Roche). Colorimetric signal was read at 450 nm. The absolute values for Aβ40 and Aβ42 are expressed as picomoles per litre of incubation medium (pM). 

*Statistical Analysis*. The two-tailed pairwise student *t*-test was employed to determine the difference between groups. In the case of data expressed as relative values, one sample *t*-test was used instead. Significance levels were denoted at *p* < 0.05 (*), *p* < 0.01 (**), and *p* < 0.001 (***). Statistical analyses were performed using Microsoft Excel (16.0, Microsoft, Redmond, WA, USA) and GraphPad Software online resources: https://www.graphpad.com/quickcalcs/oneSampleT1/ (accessed on 15 March 2024).

## 3. Results

### 3.1. HSV-1 Infection Increases Intracellular Cholesterol Levels

Previous studies conducted by our research group identified cholesterol homeostasis as one of the most disrupted processes in cell models of HSV-1 infection [14]. First, we aimed to quantify cholesterol levels in infected SK-N-MC and N2a cells using a fluorometric method. To validate this approach, we employed two cholesterol modulators known to elevate cellular cholesterol levels: U18666A and water-soluble cholesterol (Figure 1A). Upon exposure to HSV-1, both cell lines also exhibited a significant increment in cholesterol load (Figure 1B).

Another well-established method for visualizing cellular cholesterol through fluorescence microscopy involves the use of filipin, a macrolide antibiotic isolated from *S. filipinensis* that fluoresces and selectively binds to cholesterol and unesterified sterols. We employed this method to detect free cholesterol in our cell models. In uninfected cells, cholesterol predominantly localizes to the plasma membrane. However, HSV-1 infection induced a substantial accumulation of intracellular cholesterol (Figure 1C). As a positive control, cells were treated with U18666A, an inhibitor of cholesterol transport out of late endosomes/lysosomes, leading to cholesterol accumulation within endolysosomal compartments. Intriguingly, the staining pattern of HSV-1-infected cells, as well as that of U18666A-treated cells, shows differences regarding the control (Figure 1C). Additionally, the rounded morphology of filipin-positive structures suggested a vesicular nature, implying that HSV-1 infection may trigger the accumulation of cholesterol within endolysosomal compartments. These results raise the question of which cellular compartments cholesterol accumulates in in HSV-1-infected cells. With this goal in mind, the distribution pattern of filipin was examined through confocal analysis. No colocalization of filipin with any organelle marker was observed in uninfected cells (Figure 1D). When SK-N-MC cells were exposed to HSV-1, filipin staining colocalized with CD222 (a late endosome marker) and LC3 (an autophagosome marker), but no relevant colocalization of filipin with early endosomal (EEA1) or lysosomal markers (LAMP2) was observed (Figure 1E). Taken together, these findings suggest that cholesterol accumulates in late endosomal/autophagic compartments. The absence of colocalization of filipin with lysosome markers may indicate inefficient fusion between late endosomes/autophagosomes and lysosomes caused by HSV-1 infection, as previously described by our group in human neuroblastoma cells [11].

### 3.2. MβCD Reverts the Cholesterol Accumulation Induced by HSV-1

Methyl-β-cyclodextrin (MβCD) is a compound that extracts cholesterol from cellular membranes, thereby reducing its levels. Initially, the impact of MβCD on the cellular viability of SK-N-MC and N2a cells was examined. The results indicated that MβCD did not interfere with cell viability at concentrations up to 2 mM in SK-N-MC cells and 3 mM in N2a cells (Figure 2A).

To verify the effectiveness of reducing cholesterol levels with MβCD, the amount of cholesterol present in SK-N-MC and N2a cells infected with HSV-1 and subjected to increasing concentrations of the drug was quantified. The results confirmed the increase in cholesterol levels induced by the infection and revealed a dose-dependent decrease in cholesterol levels across all tested conditions with MβCD in both cell lines. Specifically, infected cells treated with concentrations of 1.5–2 mM in SK-N-MC cells and 2.5–3 mM in N2a cells showed a significant reduction in cholesterol levels, reaching levels comparable to those observed in uninfected and untreated cells (Figure 2B). Based on these findings, concentrations of 1.5–2 mM of MβCD were selected for treatments of SK-N-MC cells, while 2.5–3 mM were chosen for treatments of N2a cells in subsequent experiments.

With the aim of assessing whether MβCD treatment could reduce cholesterol accumulation in lysosomal compartments, filipin labeling experiments were performed. SK-N-MC and N2a cells were exposed to the U18666A compound, followed by MβCD treatment for this purpose. Filipin visualization through fluorescence microscopy revealed that U18666A induced intracellular cholesterol accumulation. Interestingly, MβCD resulted in a substantial decrease in filipin labeling in U18666A-treated cells (Figure 2C). 

After optimizing the treatment conditions with MβCD, the next step was to investigate whether MβCD could reduce intracellular cholesterol accumulation promoted by HSV-1. Both uninfected and HSV-1-infected SK-N-MC and N2a cells treated with MβCD were labeled with filipin. Analysis by fluorescence microscopy revealed that MβCD treatment led to a substantial reduction in cholesterol in the plasma membrane of uninfected cells. Moreover, previously observed virus-induced intracellular cholesterol accumulation was confirmed and a reduction in the intensity of intracellular filipin labeling was noted with MβCD treatment in infected cells (Figure 2D). These findings indicate that MβCD treatment results in a reduction of intracellular cholesterol accumulation in infected neuroblastoma cells.

### 3.3. MβCD Hampers HSV-1 Infection

To explore the relevance of cholesterol accumulation promoted by HSV-1, the effect of reducing cholesterol levels using the drug MβCD on various aspects related to the efficiency of infection was analyzed. Initially, we aimed to determine whether the decrease in cholesterol levels affected viral protein expression. Western blot analysis targeting the ICP4 protein (an immediately early protein essential for viral transcription and replication) and glycoprotein C (gC; a “true” late viral protein belonging to the lipid envelope) revealed a significant decrease in viral protein levels induced by MβCD in both SK-N-MC (Figure 3A) and N2a (Figure 3B) cells. Then, quantitative PCR experiments demonstrated that MβCD induced a reduction in viral DNA quantity in both cell lines, indicating its impact on the viral DNA replication process (Figure 3C). Finally, to evaluate the effect of cholesterol level reduction on the formation of infectious viral particles, we analyzed the viral titer in culture supernatants and cell lysates of MβCD-treated SK-N-MC and N2a cells. The results showed that MβCD completely blocked the production of both extracellular and intracellular infectious viral particles (Figure 3D). Collectively, these findings confirm the inhibitory effect of reduced cellular cholesterol levels on HSV-1 infection.

### 3.4. Cholesterol Participates in Post-Entry Phases of HSV-1 Infection

Several studies have identified cholesterol as essential for HSV-1 entry [21,26]. To validate this effect and check the involvement of cholesterol in entry and post-entry stages of the viral cycle in neuroblastoma cells, MβCD treatment was administered, either 1 h before (−1 hpi) or 1 h after viral adsorption (0 hpi). The impact on viral entry was assessed by monitoring the levels of viral protein ICP4 and the number of HSV-1 infected cells at early stages of infection using immunoblotting and immunofluorescence assays with a specific antibody for ICP4. ICP4 is encoded by an immediate-early gene, which is expressed prior to HSV-1 DNA replication and accumulates within the nucleus of infected cells. Assessing the ICP4 levels at early infection times serves as an indirect measure of viral entry.

Treatment with MβCD one hour before viral adsorption led to a strong reduction in both ICP4 levels (Figure 4A) and the number of HSV-1-infected cells at 5 hpi (Figure 4B). Conversely, the addition of MβCD after viral adsorption did not result in any changes in total ICP4 levels or the percentage of infected cells (Figure 4A,B). These findings highlight that, in accordance with previous data, cellular cholesterol affects HSV-1 entry in SK-N-MC and N2a neuroblastoma cells.

To assess whether cell cholesterol also influences post-entry phases of HSV-1 infection, SK-N-MC and N2a cells were exposed to HSV-1 for 2 h at 37 °C to allow viral entry. Non-internalized viral particles were subsequently inactivated by treatment with citrate buffer. The viral inactivation method involving incubation with citrate buffer at pH 3 allows the removal of all extracellular viral particles after adsorption, thereby ensuring that MβCD added to the culture medium after viral inactivation only impacts processes subsequent to viral entry. Levels of ICP4 (at 5 hpi) and gC (at 18 hpi) were analyzed using Western blot analysis. Data analysis revealed that the levels of ICP4 remained unchanged in the presence of MβCD under both experimental conditions (with or without citrate buffer), indicating that the addition of MβCD after virus adsorption does not affect viral entry (Figure 4C). The late protein gC serves as a robust marker of the late stages of infection. Interestingly, no changes in gC levels were observed upon extracellular virus inactivation, suggesting that the reduction in gC levels induced by MβCD remains consistent, irrespective of the virus inactivation state (Figure 4D).

To sum up, our findings strongly suggest that cholesterol has a pivotal role in processes subsequent to the viral entry phase. In the following experiments aimed to assess the role of cholesterol accumulation in the onset of the HSV-1-induced AD-like phenotype, MβCD was administered after viral adsorption to analyze the post-entry effects of the drug.

### 3.5. MβCD Reverts the Viral-Induced Lysosomal Alterations

To investigate the significance of cholesterol accumulation in AD-like alterations induced by HSV-1, we initially assessed its relevance in various lysosomal alterations related to HSV-1-induced lysosomal dysfunction. Previous studies from our group demonstrated that HSV-1 infection leads to an increased lysosomal content in SK-N-MC cells [14]. Consequently, we examined the impact of cholesterol reduction on lysosome load in this cell line using the fluorescent acidotropic probe LysoTracker Red (LTR).

The LTR fluorescence assay was validated with bafilomycin A1, a potent vacuolar H^+^-ATPase inhibitor known to neutralize lysosomal pH and induce a decrease in LTR fluorescence levels (Figure 5A). Quantification of LTR fluorescence levels pointed to a significant viral-induced increase in lysosomal load. However, when the effects of MβCD were examined, a substantial decrease in LTR fluorescence levels was observed in infected cells, reaching values similar to those observed in untreated and uninfected cells (Figure 5B).

To monitor the effect of cholesterol modulation with MβCD on lysosomal functionality in the SK-N-MC cell line in the presence and absence of HSV-1, the enzymatic activity of lysosomal cathepsins D/E and S was quantified using fluorogenic substrates specifically for these proteases. Previous experiments from our group revealed that HSV-1 infection decreased the activity of these hydrolases [14]. To investigate whether MβCD could reverse this effect, both uninfected and infected SK-N-MC cells were treated with MβCD. As a control, cells were exposed to bafilomycin A1, which substantially reduces cathepsin activities, validating the efficacy of the assay (Figure 5C). Enzymatic activity analysis confirmed the inhibition of cathepsin D/E and S induced by the virus. MβCD treatment did not show any effects in uninfected cells. In contrast, MβCD resulted in an increase in cathepsin enzymatic activity in HSV-1-infected cells, reaching levels similar to those of uninfected control cells (Figure 5D). 

In summary, treatment with MβCD is capable of reversing the increase in lysosomal burden and the inhibition of cathepsin activity caused by HSV-1 infection. These data suggest a role of cholesterol in the lysosomal alterations observed in infected cells.

### 3.6. MβCD Ameliorates the Neurodegenerative Phenotype Induced by HSV-1

Our group has previously demonstrated that HSV-1 and HSV-2 infection leads to intracellular accumulation of Aβ peptides and hyperphosphorylated tau [10,12,27]. Thus, our next aim was to assess the impact of cholesterol on these processes. As shown in Figure 1E, our results suggest that cholesterol accumulates in late endosomal/autophagic compartments in infected cells. Moreover, our previous research has shown that Aβ peptides accumulate in these same compartments in infected human neuroblastoma cells [12]. Considering this background, we conducted an analysis of the colocalization of cholesterol, Aβ, and phosphorylated tau (using two antibodies that specifically recognize two phosphorylated epitopes of tau protein present in AD brains: Thr205 and Ser422) in SK-N-MC cells exposed to HSV-1. Confocal microscopy analysis revealed colocalization of both isoforms of Aβ (Figure 6A) and phosphorylated tau (Figure 6B), with filipin staining in infected cells. These data suggest that the accumulation of cholesterol in endolysosomal/autophagic compartments may play a role in the intracellular accumulation of Aβ and phosphorylated tau. 

To investigate the impact of cholesterol on Aβ accumulation, we examined Aβ levels in MβCD-treated N2a cells infected with HSV-1 using confocal microscopy and ELISA assays. These experiments were conducted in N2a cells instead of SK-N-MC cells because the former secrete higher levels of Aβ, and the ELISA analysis results are more consistent and reproducible. Immunofluorescence assays revealed a decrease in the number of Aβ-positive cells in HSV-1-infected cells treated with MβCD (Figure 6C). In addition, ELISA assays were performed to determine changes in secreted Aβ levels resulting from the reduction in cholesterol amount. HSV-1 infection significantly inhibited Aβ secretion, consistent with previous findings in SK-N-MC cells [12]. In contrast to the results of immunofluorescence experiments, extracellular Aβ levels remained unchanged in MβCD-treated cells (Figure 6D).

Finally, to assess whether MβCD treatment affects the levels of phosphorylated tau protein, immunological analyses were performed in SK-N-MC cells. We found that the number of phosphorylated tau-positive cells was considerably lower in infected cells treated with MβCD, consistent with the data obtained in Aβ experiments (Figure 6E). Western blot assays also confirmed that HSV-1 induces an increase in levels of phosphorylated tau protein at the Ser422 epitope. Interestingly, we observed that the MβCD treatment leads to a decrease in levels of phosphorylated tau (Figure 6F). These findings suggest that the reduction of cholesterol levels partially reverses the accumulation of phosphorylated tau induced by the virus.

In summary, our results suggest that intracellular cholesterol accumulation may contribute to the emergence of AD markers induced by HSV-1 in neuroblastoma cells.

## 4. Discussion

HSV-1 is recognized as a risk factor in sporadic AD, as it triggers the appearance of characteristic neurodegeneration markers of the disease. Previous reports have indicated a potential link between cholesterol and both AD and viral infections [28,29]. Therefore, we hypothesized that cholesterol could mediate the interactions between HSV-1 infection and neurodegeneration. In this context, previous findings from our laboratory have implicated genes related to cholesterol metabolism and the autophagy-lysosome pathway in the process of HSV-1 infection and AD pathogenesis [20,30,31]. Thus, we aimed to assess the impact of cellular cholesterol on both infection and neurodegeneration, promoted by HSV-1 in murine and human neuroblastoma cell cultures.

The fact that viruses are able to hijack the metabolic machinery of host cells paves the way for identifying potential targets to fight against viral infections and their deleterious effects. Virtually any cellular pathway can be altered during viral infection [32]; however, lipid and cholesterol metabolism are underscored among them, given their pleiotropic nature and their role as a central node in cell physiology (reviewed in [33]). In this work, we confirm that changes in cholesterol homeostasis are one of the alterations induced by HSV-1 in our in vitro models. Indeed, we observed an increase in cholesterol levels upon infection, along with changes in its intracellular distribution pattern. Nowadays, few studies have shown connections between herpes virus infections and altered cholesterol homeostasis. HSV-1 infection has been shown to impair cholesterol metabolism, resulting in the accumulation of cholesteryl esters [34], and to alter cholesterol trafficking in human arterial smooth muscle cells [35]. To our knowledge, this is the first study to describe the accumulation of cholesterol in endolysosomal/autophagic compartments as a consequence of HSV-1 infection. This accumulation points to a likely connection between lysosomal dysfunction and alterations in cholesterol homeostasis during HSV-1 infection, supporting previous findings from our research group [14].

To investigate the role of cholesterol accumulation, we evaluated the consequences of MβCD treatment on HSV-1 infection and neurodegeneration. Cyclodextrins (CDs) are commonly used compounds that rapidly deplete cells of cholesterol. Although frequently used as molecular carriers and excipients, they have recently emerged as therapeutic agents themselves. Modified CDs have proven a broad-spectrum anti-viral activity [36]. For instance, MβCD treatment of HSV-1 particles inhibited viral entry, indicating that virion envelope cholesterol is critical for membrane fusion of viral particles [37]. In addition, CDs have been tested for clinical purposes in certain lysosome storage disorders (LSDs), such as NPC, and certain neurodegenerative diseases (reviewed in [38]). NPC is a severe congenital disease caused by mutations in the *NPC1* and *NPC2* genes, which are involved in cholesterol transport within the lysosome [39]. Commonalities between LSDs with central nervous system manifestations and AD, together with increasing evidence regarding the pivotal role of endolysosomal and autophagic dysfunction in AD, have led some authors to propose AD as an LSD itself [18].

First of all, we established the optimal conditions for MβCD treatment in our experiments to restore cholesterol levels upon infection without compromising cell viability. It is noteworthy that the degree of cholesterol reduction induced by CDs may vary considerably across different cell types, even under similar concentrations and incubation times. For instance, previous studies have shown that exposure to low concentrations of MβCD for a short duration increased cellular cholesterol in T lymphocytes from young subjects, an effect reversed by extending the exposure time [40]. These observations emphasize the importance of assessing the impact of MβCD on each specific cell line and experimental conditions. In our neuroblastoma cell-based HSV-1 infection models, an 18-h treatment with MβCD, initiated after viral adsorption, appears to be the optimal strategy for reversing virus-induced intracellular cholesterol accumulation. However, this approach has some drawbacks when studying the role of intracellular cholesterol in the AD-like phenotype caused by HSV-1. Firstly, it significantly affects the infection process. Secondly, cholesterol reduction takes place not only in endolysosomal organelles but also in all cellular membranes, raising the possibility of unwanted side effects in non-lysosomal membranes. These limitations should be borne in mind when interpreting the experimental results.

To explore whether cholesterol participates in the HSV-1 viral cycle in neuroblastoma cells, we evaluated various stages of the infection process. As an initial step, we observed a significant inhibition of viral entry with MβCD treatment. These findings align with previous reports emphasizing the relevance of cholesterol in the HSV-1 infection process, particularly in virus entry [26,41]. Furthermore, we deepened into these analyses and confirmed the participation of cholesterol in processes subsequent to the viral entry phase by adding MβCD after inactivation of non-internalized viral particles, once viral adsorption was completed. Notably, MβCD had a more pronounced effect on the formation of infectious viral particles than on the levels of viral DNA or proteins, with viral particles becoming undetectable in the presence of MβCD. These findings suggest that cholesterol may play a role in later stages of infection such as virion maturation, formation of infectious particles, or virus release from the cell. Thus, we successfully replicated the results obtained by [21], supporting our experimental hypothesis by ensuring the plausible implication of cholesterol in post-entry effects of viral infection, particularly those related to AD-like neurodegeneration.

We have previously shown that HSV-1 impairs two processes closely associated with neurodegeneration: tau phosphorylation and amyloid-beta precursor protein proteolytic processing [10,12]. In this report, we describe the accumulation of hyperphosphorylated tau and Aβ in compartments of the autophagy-lysosome pathway in infected cells. Considering the accumulation of both cholesterol and AD-like neurodegeneration markers in these compartments, it is tempting to speculate that neurodegeneration observed in HSV-1-infected cells could be linked to cholesterol accumulation. According to this hypothesis, MβCD completely restored changes related to lysosomal dysfunction, including lysosome load and cathepsin function in infected neuroblastoma cells. In this regard, MβCD is referred to as a potential treatment for LSDs and its role in restoring autophagy flux and reducing cholesterol accumulation has been confirmed in vitro [39] and in vivo [42]. Moreover, several studies have revealed that inhibition or loss of cathepsins triggers lysosomal dysfunction, leading to intracellular cholesterol and Aβ accumulation, suggesting their involvement in the appearance of neurodegeneration markers [43]. Finally, reduced lysosomal enzyme activity has been associated with AD in various reports (reviewed in [18]), underscoring the relevant role of cathepsin activity in lysosomal function and neurodegeneration avoidance.

In line with these findings, MβCD also ameliorated the alterations in phosphorylated tau and Aβ levels observed in our models. A reduction in the number of cells showing accumulation of Aβ and phosphorylated tau was observed in MβCD-treated neuroblastoma cells exposed to HSV-1. Consistent with this, quantification of intracellular phosphorylated tau revealed that MβCD treatment leads to a decrease in levels of phosphorylated tau, partially reversing the effects of HSV-1 infection. In this regard, alterations in cholesterol homeostasis have been associated with the accumulation of hyperphosphorylated tau and intracellular Aβ [44,45], and cyclodextrin treatment reduced neuroinflammation and cognitive deficits in a transgenic AD mouse model [46]. The decrease in intracellular Aβ and phosphorylated tau upon MβCD treatment described in this work could be explained by several mechanisms. First, a reduction in the number of cells in which the infection successfully progresses could be behind the alleviation of beta-amyloid and tau pathology. Second, this effect could be caused by the restoration of lysosomal function, enabling more efficient degradation of protein aberrant forms, or providing cellular protection against HSV-1 infection. Third, MβCD could promote Aβ secretion, thereby reducing intracellular Aβ accumulation. However, we have not found any effect of MβCD on the inhibition of Aβ secretion. These results suggest that MβCD treatment may interfere with viral-induced processes distinct from those involved in the Aβ secretion pathway. Considering the pleiotropic effects of HSV-1 infection, it is unlikely that a single compound is able to revert all the alterations promoted by the virus. Nevertheless, these findings could guide future experiments aiming to elucidate the molecular mechanisms behind the role of cholesterol in HSV-1-induced neurodegeneration.

In conclusion, whereas it is unlikely that efficient treatments for neurodegeneration will consist of single-target strategies, these basic research approaches are useful to progressively elucidate the complex scenarios and molecular networks involved in AD and its links with different environmental risk factors such as HSV-1 infection. Although further analyses are required to confirm these hypotheses, our proposal and the results obtained support the relevance of cholesterol in viral infection and its impact on neurodegeneration, which sheds some light on this likely bridge connecting HSV-1 with AD.

## Figures and Tables

**Figure 1 biomolecules-14-00603-f001:**
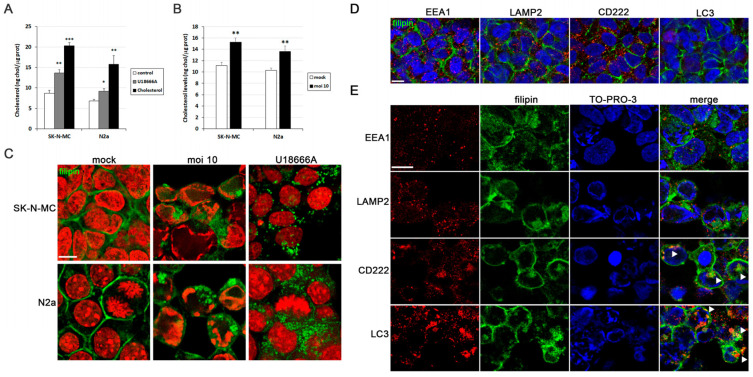
HSV-1 infection induces intracellular cholesterol accumulation. (**A**) Intracellular cholesterol levels expressed as nanograms of cholesterol per micrograms of protein in SK-N-MC and N2a neuroblastoma cells treated with U18666A and water-soluble cholesterol. (**B**) Intracellular levels of cholesterol in cultures infected with HSV-1 at a multiplicity of infection (MOI) of 10 for 18 h compared to mock-infected cells. The graph data show the mean ± SEM of at least 4 independent experiments. Significance was recorded at *p* < 0.05 (*), *p* < 0.01 (**), and *p* < 0.001 (***). (**C**) Confocal microscopy images of filipin staining (green) show cholesterol distribution patterns in SK-N-MC and N2a cells infected with HSV-1 (MOI 10) or treated with U18666A. TO-PRO-3-stained nuclei (red) are also shown. Immunofluorescence images of SK-N-MC cells uninfected (**D**) or infected with HSV-1 at MOI 10 (**E**) were obtained with filipin staining (green) and antibodies recognizing different markers (red) of early (EEA1) and late endosomes (CD222), autophagosomes (LC3), and lysosomes (LAMP2). TO-PRO-3-stained nuclei (blue) are also shown. Arrowheads in the merge panels showed the colocalization of filipin with CD222 and LC3 in HSV-1-infected cells. Scale bar: 10 µm.

**Figure 2 biomolecules-14-00603-f002:**
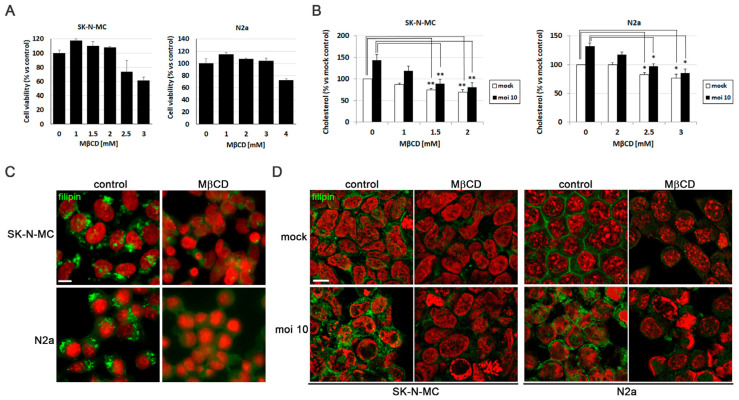
MβCD reverses the cholesterol accumulation induced by HSV-1. (**A**) Cell viability of SK-N-MC and N2a cells subjected to 18 h of treatment with different doses of MβCD was monitored using the MTT reduction assay. (**B**) Effects of different concentrations of MβCD on intracellular levels of cholesterol in infected (MOI 10) and mock-infected cells. Cholesterol levels are shown as a percentage versus mock-infected and non-treated cultures. The graph data show the mean ± SEM of at least 3 independent experiments. Significance was recorded at *p* < 0.05 (*) and *p* < 0.01 (**). (**C**,**D**) The effect of MβCD on cholesterol distribution patterns in SK-N-MC and N2a cells treated with U18666A (**C**) or infected with HSV-1 at MOI 10 for 18 h (**D**) was tested. Fluorescence microscopy images show filipin staining in green and cellular nuclei stained with TO-PRO-3 in red. Scale bar: 10 µm.

**Figure 3 biomolecules-14-00603-f003:**
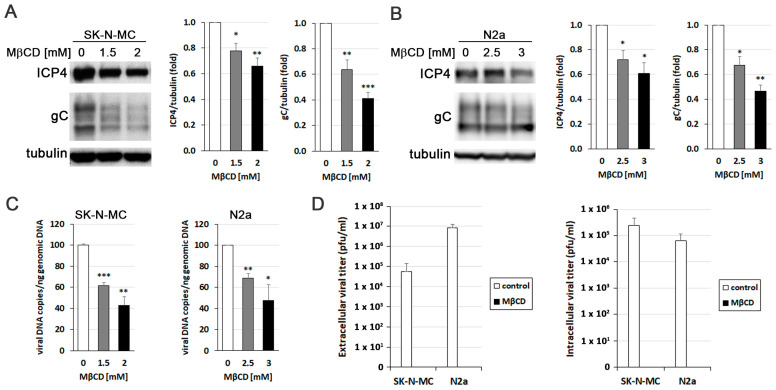
MβCD hampers the efficiency of HSV-1 infection. The impact of MβCD treatment on the levels of viral proteins ICP4 and glycoprotein C (gC) was determined through Western blot analysis in SK-N-MC (**A**) and N2a (**B**) cells exposed to HSV-1 at MOI 10 for 18 h. A tubulin blot is shown to ensure equal protein load. In all graphs, data represent the mean ± SEM of 3 experiments. (**C**) The effect of MβCD on viral replication was evaluated by quantifying viral DNA copies in both neuroblastoma cell lines. The graph data show the mean ± SEM of 4 independent experiments. Significance was recorded at *p* < 0.05 (*), *p* < 0.01 (**), and *p* < 0.001 (***). (**D**) Intracellular and extracellular viral titers were determined by plaque assays in SK-N-MC and N2a cells infected with HSV-1 at MOI 10 for 18 h and subsequently exposed to MβCD (2 mM for SK-N-MC cells and 3 mM for N2a cells). The graph data show the mean ± SD from 3 (extracellular titer) or 2 (intracellular titer) independent experiments. Original images of (**A,B**) can be found in Appendix A.

**Figure 4 biomolecules-14-00603-f004:**
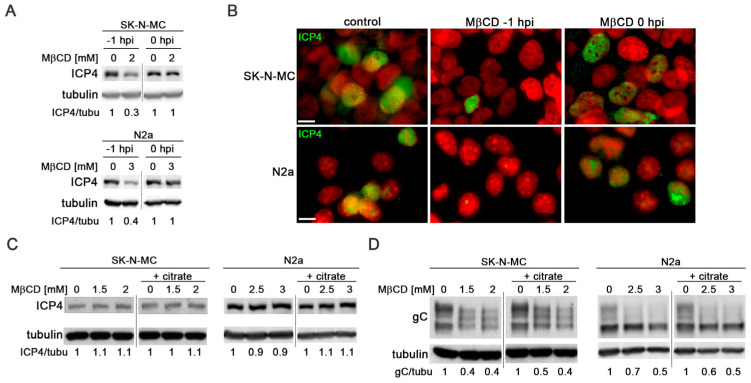
MβCD affects the post-entry phases of HSV-1 infection. (**A**) The effects of MβCD treatment at different times (1 h before (−1 hpi) or just after (0 hpi) viral adsorption) on ICP4 levels were assessed through Western blot analysis in SK-N-MC and N2a cells infected with HSV-1 at MOI 10 for 5 h. (**B**) Immunofluorescence images of ICP4 (green) in neuroblastoma cells infected with HSV-1 at MOI 10 for 5 h show the effects of MβCD added either before (−1 hpi) or after (0 hpi) viral adsorption. DAPI-stained nuclei are also shown (red). Scale bar: 10 µm. (**C**,**D**) SK-N-MC and N2a cells infected with HSV-1 (MOI 10) were treated with citrate buffer (pH 3) to inactivate the non-internalized virus. Cells were then treated with different concentrations of MβCD. Levels of ICP4 (**C**) and gC (**D**) viral proteins were monitored at 5 and 18 hpi, respectively, by Western blot analysis. In all panels, a tubulin blot is shown to ensure equal protein load, and the ratio of viral proteins to tubulin, estimated by densitometric analysis, is shown below the blots. In (**A**,**C**,**D**), the vertical black lines indicate that the blots have been spliced. The spliced fragments come from the same original image. Original images of (**A**,**C**,**D**) can be found in Appendix A.

**Figure 5 biomolecules-14-00603-f005:**
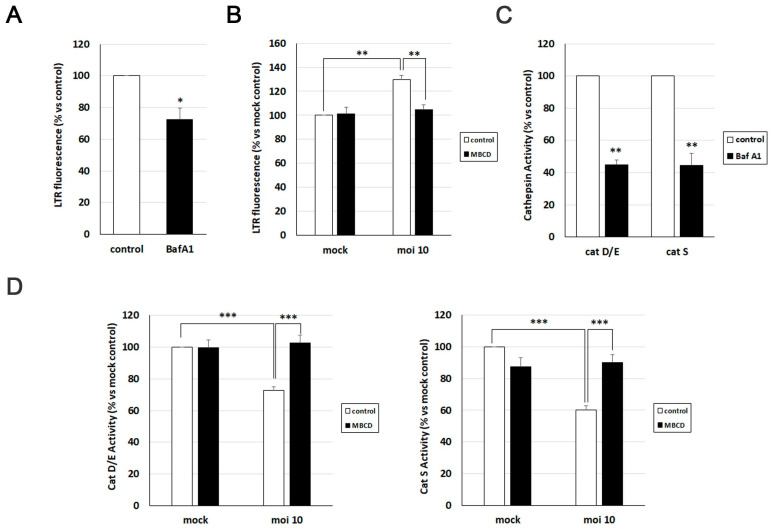
MβCD treatment reverses the lysosomal alterations induced by HSV-1. (**A**,**B**) Assessment of lysosomal load by measurement of LysoTracker Red (LTR) fluorescence in SK-N-MC cells treated with 0.1 µM bafilomycin A1 (**A**) or infected with HSV-1 at MOI 10 and treated with 2 mM MβCD for 18 h (**B**). (**C**,**D**) The relative enzymatic activities of cathepsins D/E and S were evaluated in SK-N-MC cells treated with 0.1 µM bafilomycin A1 (**C**) or infected with HSV-1 at MOI 10 and treated with 2 mM MβCD for 18 h (**D**). The graph data show the mean ± SEM from at least 4 independent experiments. Significance was recorded at *p* < 0.05 (*), *p* < 0.01 (**), and *p* < 0.001 (***).

**Figure 6 biomolecules-14-00603-f006:**
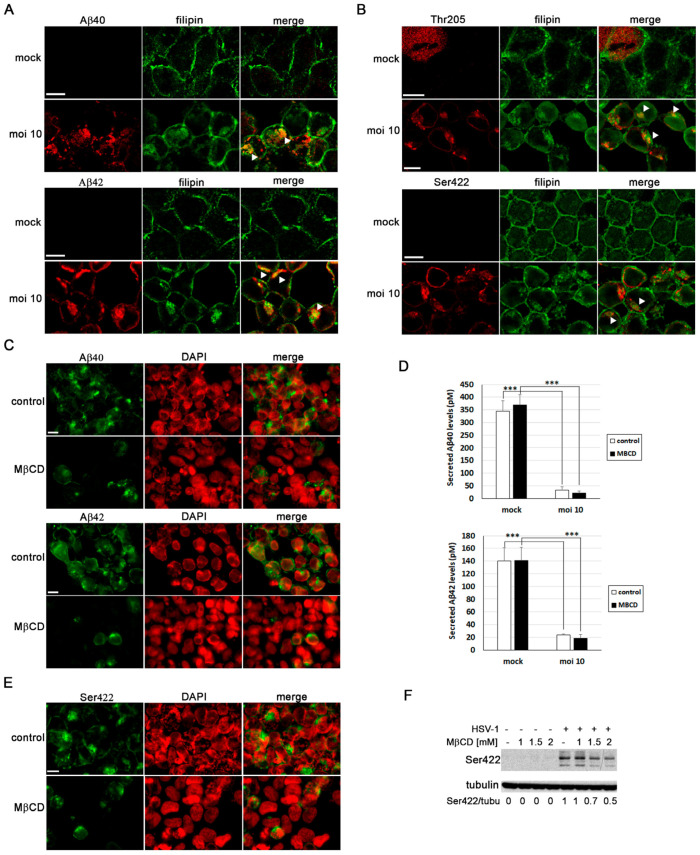
MβCD ameliorates the neurodegenerative phenotype associated with viral infection. (**A**,**B**) Immunofluorescence images of SK-N-MC cells uninfected (mock) or infected with HSV-1 at MOI 10 for 18 h were obtained with filipin staining and antibodies recognizing Aβ peptides (**A**) and phosphorylated tau at epitopes Thr205 and Ser422 (**B**). Arrowheads in the merge panels showed the colocalization of filipin with Aβ and phosphorylated tau in HSV-1-infected cells. Scale bar: 10 µm. (**C**) N2a cells were infected with HSV-1 at MOI10 for 18 h and treated with 3 mM MβCD. Immunofluorescence images show Aβ40 and Aβ42 staining in green and cellular nuclei stained with DAPI in red. Scale bar: 10 µm. (**D**) Quantitative analysis of secreted Aβ levels by ELISA. N2a cells were infected with HSV-1 (MOI10) for 18 h and treated with 3 mM MβCD. The graph data show the mean ± SEM of 6 independent experiments. Significance was recorded at *p* < 0.001 (***). (**E**) SK-N-MC cells were infected with HSV-1 at MOI 10 for 18 h and treated with 2 mM MβCD. Phosphorylation of the Thr205 epitope of tau was analyzed by immunofluorescence. DAPI-stained nuclei are also shown (red). Scale bar: 10 µm. (**F**) SK-N-MC cells were infected with HSV-1 at MOI 10 for 18 h and treated with different concentrations of MβCD. Levels of Ser422-phosphorylated tau were monitored by Western blot analysis. A tubulin blot is shown to ensure equal protein load. The ratio of phosphorylated tau to tubulin, obtained by densitometric analysis, is shown below the blots. Original images of (**F**) can be found in Appendix A.

**Table 1 biomolecules-14-00603-t001:** Summary of antibodies employed in Western blot (WB) and immunofluorescence (IF) experiments.

Primary Antibodies		Species	Dilution	Reference
HSV-1 infection	ICP4	Mouse	1/1000 (WB) 1/100 (IF)	Abcam ab6514
	gC	Mouse	1/3000 (WB) 1/300 (IF)	Abcam ab6509
Neurodegeneration markers	Aβ40	Rabbit	1/100 (IF)	Invitrogen 44348A
	Aβ42	Rabbit	1/100 (IF)	Invitrogen 44-344
	p-Tau Thr205	Rabbit	1/250 (WB) 1/50 (IF)	Invitrogen 44-738G
	p-Tau Ser422	Rabbit	1/250 (WB) 1/50 (IF)	Invitrogen 44-764G
Autophagy-lysosomal pathway	EEA1	Mouse	1/1000 (WB) 1/100 (IF)	BD Biosciences 610457
	Human LAMP2	Mouse	1/1000 (WB) 1/50 (IF)	DSHB H4B4
	CD222	Mouse	1/100 (IF)	BioLegend 315902
	LC3B	Rabbit	1/500 (WB) 1/100 (IF)	Sigma L7543
Housekeeping protein	α-Tubulin	Mouse	1/10,000 (WB)	Sigma T5168
**Secondary Antibodies**		**Species**	**Dilution**	**Reference**
Mouse-POD		Horse	1/2,5000 (WB)	Vector PI-2000
Rabbit-POD		Goat	1/25,000 (WB)	Nordic GAR/IgG (H + L)/PO
Alexa-555 anti-mouse		Goat	1/1000 (IF)	Thermo Fisher A-21137
Alexa-488 anti-rabbit		Donkey	1/1000 (IF)	Thermo Fisher A-21206

## Data Availability

The original contributions presented in the study are included in the article/Appendix A, further inquiries can be directed to the corresponding author/s.

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
