# Peer review of "Cholesterol Modulation Attenuates the AD-like Phenotype Induced by Herpes Simplex Virus Type 1 Infection"

_biomolecules, 2024, doi:10.3390/biom14050603_

Round 1
Reviewer 1 Report
Comments and Suggestions for Authors
Very well-written and thorough investigation of the interplay between HSV and cholesterol in viral effects on 2 cell lines of neuronal background, one human and the other murine. Figures are clear and discussion is appropriate.
A few issues:
1) In a paper focusing on cholesterol effects in relation to AD, ApoE genotype in the cell model is relevant. While ApoE alleles have been documented for some cell lines (SK-N-SH are ApoE3/E3), the SK-N-MC ApoE genotype is not in the literature. You can easily determine this. It would be interesting to see any modulatory effects of ApoE isoform on your findings.
2) Beta-cyclodextrins are known for their anti-viral activity in general. Please add brief discussion of this (examples: Wudiri GA, Schneider SM, Nicola AV. Herpes Simplex Virus 1 Envelope Cholesterol Facilitates Membrane Fusion. Front Microbiol. 2017;8:2383. Published 2017 Dec 6. doi:10.3389/fmicb.2017.02383 and Jones ST, Cagno V, Janeček M, et al. Modified cyclodextrins as broad-spectrum antivirals. Sci Adv. 2020;6(5):eaax9318. Published 2020 Jan 29. doi:10.1126/sciadv.aax9318).
3) Line 134: Please correct “culured” to “cultured”
Reviewer 2 Report
Comments and Suggestions for Authors
The authors investigated the role of cellular cholesterol in HSV-1-induced neurodegeneration in neuronal cell models using state-of-the-art methodologies. Through these studies they found that HSV-1 infection triggered cholesterol accumulation in specific subcellular compartments. Decreasing cholesterol levels using MβCD interfered with HSV-1 infection as well as HSV-1-induced neurodegeneration.
Overall, this work identified novel aspects of cholesterol homeostasis during HSV-1 infection in neuroblastoma cell lines. The methodologies used are adequate and the findings are of interest to a broader audience. Below are a few considerations which, I believe, will increase the quality of this work even further.
Major points
· Since HSV-1 contains cholesterol and a moi of 10 is very high, this might affect the cellular cholesterol measurements (Fig. 1B). An appropriate control (e.g. inactivated virus) could rule out a contribution of HSV-1 cholesterol.
· It is not clear whether HSV-1 came in contact with MβCD during the experiments. This is of relevance since it has been suggested that HSV-1 cholesterol is important in the entry process.
· In Fig. 1D, it would be informative to present the immunostainings for the control as well. In addition, a cytosol, plasma membrane, or at least nuclear stain would make it easier to appreciate the subcellular localization. Some of the colocalizations are not very easy to see and thus I would recommend insets with higher resolution as well as quantifications.
· Why is there only data from one representative experiment shown in Fig. 3D? The data presented is not very convincing.
Minor points
· Some of the scale bars seem to have wrong dimensions, please double-check them.
· In the first paragraph of section 3.1.2. it is confusing to the reader that the “AD-like phenotype” is mentioned.
· The Amplex red assay detects both, cholesterol and cholesteryl esters. Since the finding that HSV-1 induced cholesterol accumulation is central to the work, it might be worth using an alternative method to quantify cholesterol and cholesteryl esters.
· The use of “significant” should only be used where significance is shown, e.g. “No significant colocalization with filipin” in the text relating to Fig. 1D.
· In the text relating to Fig. 1C, it is stated: “Intriguingly, the staining pattern of HSV-1-infected cells closely resembled that observed in U18666A-treated cells”. The staining pattern is quite different and hence “closely resemble” does not reflect the data. Please rephrase.
· Generally, I recommend avoiding the use of green and red in combination to visualize immunostainings.
· In Fig. 2B, it is unclear among which groups the statistical analyses were performed. I recommend using a similar approach as in Fig. 5B.
· I believe Figs. 6A+B would be easier to read if the mock control was presented in the same layout, like it has been done for Figs. 6C.
· In the discussion, the Aβ secretion data (Fig. 6D) should be discussed.
Comments on the Quality of English LanguageMinor language editing will improve the quaility of this manuscript.
Round 2
Reviewer 2 Report
Comments and Suggestions for Authors
The authors addressed all my concerns and I have no more comments.